# Smoking, Lung Cancer Stage, and Prognostic Factors—Findings from the National Lung Screening Trial

**DOI:** 10.3390/ijerph21040400

**Published:** 2024-03-26

**Authors:** Junjia Zhu, Steven Branstetter, Philip Lazarus, Joshua E. Muscat

**Affiliations:** 1Department of Public Health Sciences, Penn State College of Medicine, Hershey, PA 17033, USA; jzhu2@pennstatehealth.psu.edu; 2Department of Biobehavioral Heath, Penn State University, University Park, PA 16802, USA; sab57@psu.edu; 3Department of Pharmaceutical Sciences, Washington State University, Spokane, WA 99202, USA; phil.lazarus@wsu.edu

**Keywords:** lung cancer, nicotine dependence, screening, survival, time to first cigarette, cancer stage

## Abstract

Background: Low-dose computed tomography (LDCT) increases the early detection of lung cancer. Identifying modifiable behaviors that may affect tumor progression in LDCT-detected patients increases the likelihood of long-term survival and a good quality of life. Methods: We examined cigarette smoking behaviors on lung cancer stage, progression, and survival in 299 ever-smoking patients with low-dose CT-detected tumors from the National Lung Screening Trial. Univariate and multivariate Cox models were used to estimate the hazard ratio (HR) for smoking variables on survival time. Results: Current vs. former smokers and early morning smokers (≤5 min after waking, i.e., time to first cigarette (TTFC) ≤ 5 min) had more advanced-stage lung cancer. The adjusted HR for current vs. former smokers was 1.3 (95% confidence interval [CI] 0.911–1.98, *p* = 0.136) for overall survival (OS) and 1.3 (0.893–1.87, *p* = 0.1736) for progression-free survival (PFS). The univariate hazard ratios for TTFC ≤ 5 min vs. >5 min were 1.56 (1.1–2.2, *p* = 0.013) for OS and 1.53 (1.1–2.12, *p* = 0.01) for PFS. Among current smokers, the corresponding HRs for early TTFC were 1.78 (1.16–2.74, *p* = 0.0088) and 1.95 (1.29–2.95, *p* = 0.0016) for OS and PFS, respectively. In causal mediation analysis, the TTFC effect on survival time was mediated entirely through lung cancer stage. Conclusion: The current findings indicate smoking behaviors at diagnosis may affect lung cancer stage and prognosis.

## 1. Introduction

Most lung cancer patients are diagnosed with advanced disease and have a poor prognosis. Since 2004, approximately 70% of lung cancers are Stage III or IV in the Surveillance, Epidemiology, and End Results (SEER) Program [1]. A late stage of disease is the primary determinant of lung cancer progression and survival. Smoking behaviors at the time of diagnosis may also affect disease progression. In a meta-analysis of 21 studies, the overall survival risk in smokers was improved in patients who reported recent smoking cessation vs. current smoking and this was consistent across all lung cancer cell types (hazard ratio (HR) = 0.71, 95% confidence interval (CI) 0.64–0.80) [2]. Since these studies were published, annual low-dose CT screening for lung cancer has become increasingly available nationally and is utilized for smokers that meet the screening criteria. The United States Preventive Services Task Force recommends annual screening for adults aged 50 to 80 years with at least 20 pack-years of cigarette smoking. Former smokers must have quit smoking within the past 15 years to be eligible. In the National Lung Screening Trial, only 35% of LDCT-screened participants were diagnosed with stage III or IV disease [3]. The early detection of lung cancer using low-dose computed tomography offers the opportunity to reduce lung cancer mortality [4]. Consequently, the effect of smoking status in newly diagnosed LDCT-detected lung cancer patients on improved survival might be substantial, given their overall lower stage of disease. In the current study, we investigated whether smoking behaviors affect the extent of disease at diagnosis and subsequent progression of lung cancer in LDCT-detected screening. In addition to current smoking status, other modifiable smoking behaviors may also affect the prognosis. Among ever-smokers, an early time to first cigarette (TTFC) is an independent risk factor that further doubles the lung cancer incidence and mortality risk [5,6,7,8,9,10]. It is unclear what the mechanisms are that explain how smoking behaviors might affect lung cancer progression and mortality. Smoking factors may affect the extent and spread of the tumor, since cigarette smoke contains numerous compounds that affect uncontrolled cell proliferation which are thought to be factors in tumor growth and spread [11]. Clinically, this is characterized by the stage of disease, which can be classified based on the tumor size and any spread to adjacent lymph nodes. An effect of smoking behaviors and smoking dose on lung cancer progression may be due to increased cancer growth and cell proliferation. The effects of TTFC on lung cancer stage and progression have not been studied. Similarly, the effects of current smoking in LDCT-detected lung cancer patients have not been previously reported. To better understand how smoking behaviors (e.g., TTFC, smoking status) might affect lung cancer stage, progression, and mortality, we conducted a secondary analysis of a large multi-center study of lung cancer screening.

## 2. Materials and Methods

### 2.1. Participants

The National Lung Screening Trial (NLST) was conducted in 33 U.S. sites. It enrolled 53,456 current or former smokers with a history of at least 30 pack-years starting from 2002–2004 [4]. Participants were randomized to either the low-dose computerized tomography arm or standard chest radiography arm in a 1:1 ratio. The randomization was stratified according to age, sex, and screening center. The objective of the NLST was to determine if low-dose computerized tomography (LDCT) reduces mortality from lung cancer compared to standard chest X-ray. The NLST included the Lung Screening Study (LSS) network and the American College of Radiology Imaging Network (ACRIN). A total of 18,842 consenting participants were administrated by ACRIN, which had a supplemental baseline questionnaire on smoking and nicotine addiction behaviors including the time to first cigarette. NLST-ACRIN issued a data sharing agreement with us for 14,028 ACRIN participants. ACRIN reserved ~25% of its data for verification purposes. In our analysis there were 7038 ACRIN participants who were randomized to the LDCT arm, and 299 of these had a confirmed diagnosis of lung cancer. Participants were followed for up to seven years until 31 December 2009. The study was approved by the Penn State Hershey Medical Center Institutional Review Board (#1885).

### 2.2. Outcome Variables

Clinical data were obtained from pathology and cancer-staging reports. Lung cancer disease stage was coded using the Cancer Staging Manual of the American Joint Committee on Cancer (AJCC) 7th edition [12]. Stage was grouped into “early” and “advanced” stage (early stage: 1A-3A; advanced stage: 3B-4). Two types of survival time were measured: overall survival (OS) and progression-free survival (PFS). The overall survival time is defined as the length of time from the date of cancer diagnosis to the date of death or to the date of last follow-up. The PFS time is defined as the length of time from the date of diagnosis to first disease progression (such as enlargement of the original tumor, new metastasis to lymph nodes or other organ site not included in the original tumor staging, or disease recurrence). Death was not considered as a disease progression endpoint for PFS. Note that the exact dates of diagnosis were not provided by NLST due to the protection of patients’ confidentiality. However, the OS and PFS time could still be determined by calculating the differences between variables that quantified the time from randomization to death or progression, and time from randomization to cancer diagnosis.

### 2.3. Statistical Methods

Descriptive statistics were generated to describe patient characteristics including age, sex, race, education (high school or less, some college, college degree), marital status, personal and family history of cancer (yes/no), and body mass index (BMI). BMI was categorized into four categories including underweight (BMI < 18.5), normal (18.5 ≤ BMI < 25), overweight (25 ≤ BMI < 30), and obese (BMI ≥ 30). The histology and grade were coded in the NLST using the International Classification of Diseases for Oncology, 3rd Edition (ICD-O-3) [13]. Cell type was grouped into small cell, and non-small cell lung cancer.

Smoking measures for the current analysis included smoking status (current, former), years of smoking and cigarettes per day, and the time to first cigarette, which had four response categories (≤5 min, 5–29 min, 30–59 min, ≥60 min). These are commonly categorized into two categories (within 5 min vs. longer). The time to first cigarette is considered the best single indicator of nicotine dependence and is an item on the 6-item Fagerstrom Test for Nicotine Dependence (FTND) [14].

The data in this study were described using means and proportions. Cox proportional hazard regression models were conducted to determine the relationship between smoking variables and survival time. The direction and magnitude of the association were quantified using the estimated hazard ratios and their 95% confidence intervals. Multivariate Cox proportional-hazards models were subsequently conducted for smoking variables, while controlling for cancer stage and other confounders [15]. Where there was evidence of high multicollinearity (e.g., tumor size with cancer stage), we limited the variable selection to stage for the cancer diagnosis variables. Other covariates considered for the models included sex and BMI.

As the TTFC showed an effect on prognosis, we explored this relationship further using Kaplan–Meier survival curves to visualize the overall and PFS survival probability by bivariate TTFC classifications, and by the initial four level classification of TTFC. The differences in curves were compared statistically using the log-rank test.

We then used causal mediation analyses to examine the mediation effect of lung cancer stage (advanced vs. early) on the pathway between the TTFC exposure and survival. The causal mediation method consisted of determining the total effect of the exposure on survival, and determining and comparing the direct effect of the exposure on OS and PFS, to the indirect effect of the exposure on survival through the mediator (lung cancer stage), while controlling for potential confounders. For the total effect, we first developed a parametric survival regression model (using “survreg” function in R) of the survival outcome against the exposure. To determine the effect of the mediator on the dependent variable, we used the exponential parametric survival regression model (using “survreg” function in R) of the survival outcome (OS and PFS) against the exposure and the mediator, while controlling for the confounders. The statistical significance of the exposure–mediator interaction was tested using the test.TMint function [16].

All analyses were performed using statistical analysis package SAS version 9.4 (SAS Institute, Cary, NC, USA) and statistical programming language R version 4.3.2 (The R Foundation for Statistical Computing). R package “mediation” (version 4.5.0) was used for statistical mediation analysis, which uses simulations to calculate the total effect, average direct effects (ADEs), and average direct causal mediation effects (ACMEs). The user can select the number of simulations. We selected 50,000 simulations for the current analysis. The method calculates quasi-Bayesian 95% confidence intervals around the parameter estimates. All tests were two-sided and the statistical significance level used was 0.05.

## 3. Results

### 3.1. Study Sample

The study included 299 newly diagnosed lung-cancer patients in the low-dose CT arm. Table 1 shows the descriptive statistics for their demographics, smoking behaviors, measures of nicotine dependence, and clinicopathologic factors. Only 3.7% of participants had a cancer prior to the trial, and 23.4% had a family history of lung cancer. For tumor characteristics, 52.4% were grade 3 or 4 lung cancer (Table 1). The mean tumor size was 26.3 mm. Stage information was missing for 21 participants. Of the 278 participants with staging data, 69.4% were early stage lung cancers and 30.6% were advanced-stage lung cancers. The cancer stage was highly associated with other pathologic features. The mean lesion size was 21.8 mm (±15.4) in early stage cancers and 44.6 mm (±27.5) in late stage cancers. Eighty-nine percent of advanced-stage cancers were high-grade tumors, compared to 44% of early stage cancers. Approximately 89% of all subjects had non-small cell lung cancer and 11% had small cell lung cancer.

Sixty-nine percent of advanced-stage cancer patients were current smokers vs. 55% for early-stage cancer patients (*p* < 0.05). An early time to first cigarette (≤5 min) vs. >5 min was associated with advanced-stage lung cancer (40.5% vs. 23.5%, *p* = 0.004).

### 3.2. Survival Time Analysis

The mean follow-up time was 2.9 years (±2.2, ranging 0–7.5 years) for OS and 2.6 years (±2.3, ranging 0–7.5 years) for PFS. The lung cancer stage and pathology measures were associated with OS and PFS. The hazard ratio for advanced vs. early stage lung cancer was 7.2 (95% CI 4.94–10.6) for OS and 7.4 (95% CI 5.18–10.7) for PFS (Table 2). The hazard ratio for small cell vs. non-small cell type was 3.6 (95% CI 2.32–5.71) for OS and 3.5 (95% CI 2.24–5.35) for PFS. Current smokers had a significantly lower OS than former smokers (*p* = 0.0373), but the association was not significant for PFS. Years of cigarette smoking was not associated with survival time. An earlier time to first cigarette was associated with lower survival. Using four categories of the exposure variable, both OS and PFS was lowest in the ≤5 min category and highest in the ≥60 min category (Figure 1). The hazard ratios for ≤5 min vs. >5 min were 1.56 (95% CI 1.1–2.2) for overall survival (*p* = 0.013) and 1.53 (95% CI 1.1–2.12) for progression-free survival (*p* = 0.01) (Table 2). The hazard ratios for ≤5 min vs. >1 h were 2.4 (0.859–6.49) and 2.8 (1.02–7.63) for OS and PFS, respectively.

Other predictors of lower OS and PFS included male sex (*p*-value = 0.0231 and 0.0488, respectively; Table 2). Obesity was associated with significantly lower PFS than normal weight (*p* = 0.0191). A similar association of obesity on OS was found but this was not statistically significant. Other FTND items such as smoking when ill had no association. There was also no association with a family history of lung cancer.

### 3.3. Multivariate Results

The results of the multivariate survival models are shown in Table 3. For the lung cancer stage, the results were similar to the unadjusted model. The hazard ratio of advanced vs. early stage lung cancer was 7.2 (95% CI 4.81–10.7) for OS and 7.5 (95% CI 5.11–11.0) for PFS. The adjusted association between time to first cigarette ≤ 5 min and survival time was no longer significant with the addition of lung cancer stage in the model, and the association between current smoking and OS was also no longer significant. An additional model that further incorporated cigarettes per day did not change the results. The BMI was only included in the multivariate model for PFS, since the BMI was not associated with OS in the bivariate analysis. Compared to normal weight individuals, obesity was associated with a lower PFS (*p* = 0.0209).

### 3.4. Mediation Analysis

The theoretical framework for causal mediation analysis for TTFC on both OS and PFS is shown in Figure 2. The results of the analysis are shown in Table 4. There was no interaction effect between the time to first cigarette and stage for OS (*p* = 0.439) or PFS (*p* = 0.85). The ACMEs estimates for OS and PFS were statistically significantly different from zero, but the average direct and total effects were not significant.

## 4. Discussion

The current study shows that smoking behaviors (e.g., TTFC, smoking status) are associated with lung cancer stage, progression, and mortality. These associations may be either direct or mediated through the stage depending on the relationship. In order to examine these relationships, it was necessary to study them in a cohort with comprehensive data on smoking behaviors, lung cancer pathology, and survival outcomes. When diagnosed using a chest X-ray, approximately 65% of lung cancers were stage III or IV. In the NLST, it was 31% [17]. The NLST therefore had relatively greater numbers of early stage cases that facilitated the analysis. The NLST is also representative and consistent with population-based data such as SEER in that a late stage of disease is the main determinant of survival time.

Current smokers had a lower OS and PFS than former smokers. The finding was not significant after adjustment for stage and other factors, but was consistent with a pooled analysis of the Clinical Outcomes Studies (COS) of the International Lung Cancer Consortium (former smokers vs. current smokers; 0.88 (95% CI 0.86–0.91). Similarly, in a meta-analysis, currently smoking lung cancer patients had poorer survival times than former smokers [2] (HR = 1.3 in NLST vs. 1.4 in a meta-analysis). Still, the lack of statistical significance in the NLST indicates that the findings in an LDCT-screened population need confirmation in other studies. Delivering smoking cessation treatment with LDCT screening is a recommended practice for LDCT screening that potentially reduces the future risk in current smokers with no detectable lesions [18]. The benefits of cessation in LDCT screenees may extend beyond just the risk of lung cancer in healthy smokers. These data indicate that cessation might also have benefits of improved outcomes for those LDCT patients with lung cancer.

Our study also shows that an early time to first cigarette is associated with lower OS and PFS in our univariate analysis. Previous findings from the National Lung Screening Trial (NLST) found that that an early time to first cigarette increases the lung cancer mortality rate in this population [19]. In our analysis, the effects of early TTFC on poorer prognosis in univariate analysis were not observed in the multivariate analysis. The mediation models showed complete mediation, indicating that the TTFC effect on survival time was mediated entirely through lung cancer stage. The mechanism by which early TTFC results in a more advanced clinical stage, and the subsequent effect on survival time, is uncertain. Traditionally, TTFC is a behavioral measure of tobacco dependence. It is also an established marker of nicotine and tobacco smoke uptake. Levels of nicotine and tobacco smoke carcinogen metabolites are approximately twice as high in smokers who take their first cigarette immediately after waking compared to waiting an hour in nationally representative and other studies [20,21,22,23]. An early time to first cigarette likely increases exposure to the many genotoxins and co-carcinogens in tobacco smoke, which affect the tumor cell cycle, progression, and angiogenesis [24,25], which are critical factors in the growth of solid tumors and their spread [11]. In humans, the clinical effect of increased cancer spread is determined by the stage of disease.

An early TTFC has been suggested as an additional criterion for LDCT screening eligibility [26]. The current study supports this idea, not just for identifying smokers at the highest risk of lung cancer, but those at the highest risk for poorer progression if diagnosed.

Although it is well established that smoking causes lung cancer, there are limited data on smoking and other factors on lung cancer stage at diagnosis. In one cohort, socioeconomic factors, health behaviors, and medical history including BMI were not associated with stage of lung cancer [27]. Smokers with socioeconomic deprivation may be less likely to recognize lung cancer symptoms or seek a diagnosis, and are diagnosed at a later stage of disease. A meta-analysis of studies examining low levels of income or education, however, showed no overall risk with either the interval from symptoms to diagnosis, time from diagnosis to treatment, or lung cancer stage [28]. In the NLST, eligible subjects were asymptomatic and any possible effect of lower education on delayed treatment would not affect the current findings.

The strengths of the current study include its multi-center locations, very-high-quality control, assessment of nicotine dependence behaviors, and longitudinal design. The methods for this NLST were previously described in detail [29]. Although the literature on cigarette smoking and lung cancer risk spans many decades, there are almost no reports on smoking factors and stage of lung cancer. The simultaneous collection of smoking exposure information, lung cancer stage, and progression outcomes may be unique to the NLST design. Another possible limitation is that some lung tumors are difficult to discriminate between a primary lung cancer from a secondary metastasis to the lung [30]. This misclassification may affect the validity of prognostic factors in statistical models. However, this misclassification would not occur in the NLST analysis, where early lung lesions detected using screening are followed prospectively to determine the development of primary lung cancer. We found that men had a poorer survival time than women, which has been documented in other populations [31]. Age was not a prognostic factor, which is also consistent with previous findings [32]. Obesity was associated with a worse progression in the NLST. Other studies have shown a better prognosis with obesity, although it has been suggested that the reasons for this could be due to limitations in the study methodology, bias, and misclassification [33,34]. Although causal mediation analysis is commonly used in cross-sectional studies, its interpretability is more well-grounded in a prospective design where a temporal sequence is well established [35]. The interpretation of mediation analysis assumes a lack of confounders in the mediator–outcome relationship. We found no differences in the unadjusted and adjusted hazard ratios for stage on survival.

In regard to the limitations, the NLST included only persons with a 30+ pack-years smoking history, which is not representative of all smokers. Smokers who quit more than 15 years ago or were outside the age range of 55–74 were not eligible for screening. In addition, the cancer stage was missing for 21 participants. This could potentially introduce a bias if missing data were non-random across measures of smoking. There were also few black participants, or people of other races and participants with Hispanic ethnicity. In the Surveillance, Epidemiology and End Results (SEER) database, black and Asian patients have a less rapid progression of lung cancer than white patients [36]. While causal mediation analysis revealed a likely pathway for the observed TTFC findings, it is possible that there are other mediators or co-mediators that could affect these relationships. Another consideration when interpreting the current results is that we did not have post-diagnosis smoking information. Quitting smoking at around the time of diagnosis improved the lung cancer survival rate in participants followed up for several decades [2]. The follow-up time was relatively short in the NLST, so we could not assess whether this lack of information may have biased the findings.

## 5. Conclusions

It is established that smoking causes lung cancer. Here, we show that smoking behaviors also affect the lung cancer stage. A shorter lung cancer survival time with early morning smoking can be attributed to the effects of TTFC on the later stage of disease.

## Figures and Tables

**Figure 1 ijerph-21-00400-f001:**
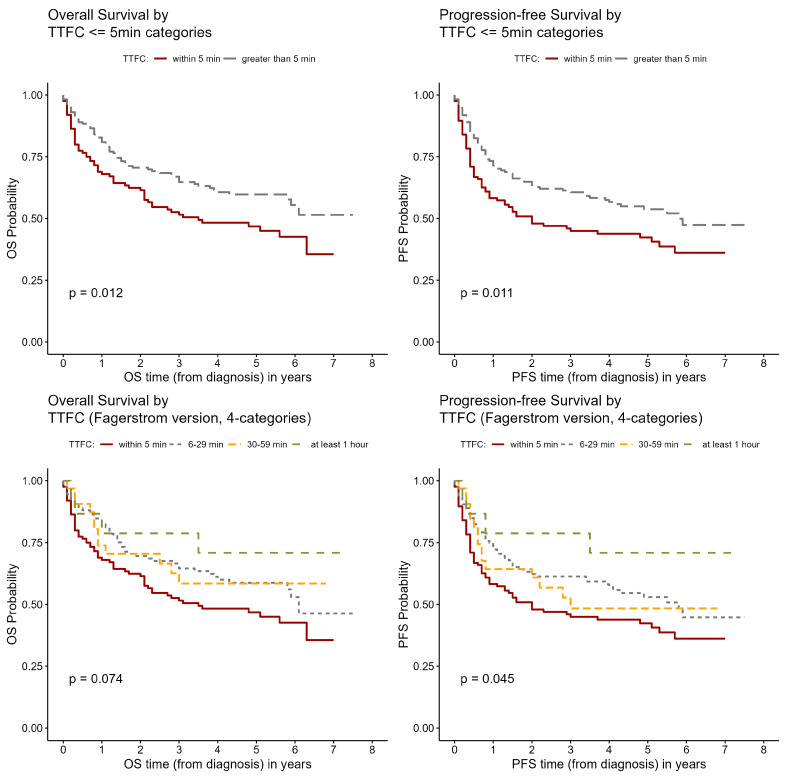
Kaplan–Meier survival curves of OS/PFS against time to first cigarette (TTFC).

**Figure 2 ijerph-21-00400-f002:**
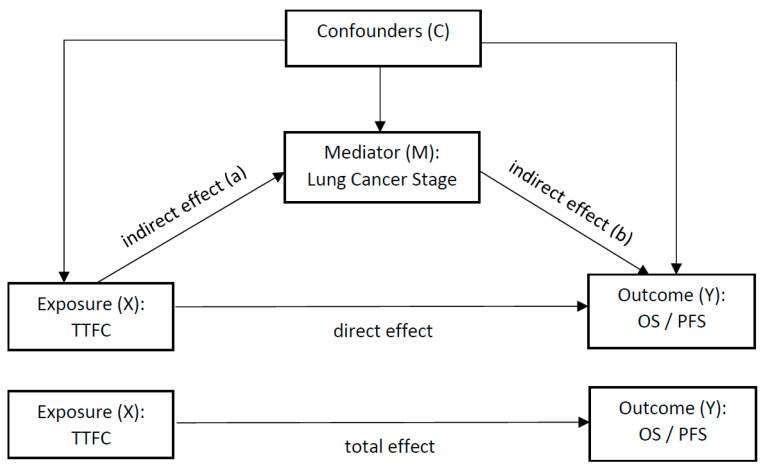
Mediation analysis pathway.

**Table 1 ijerph-21-00400-t001:** Characteristics of the NLST study sample (*n* = 299).

Variable	Statistics Type or Categories	Summarization
Age	Mean (SD)	63.7 (5.6)
Sex	Female	124 (41.5%)
	Male	175 (58.5%)
Race	White	280 (93.6%)
	Black	14 (4.7%)
	Other	5 (1.7%)
Ethnicity	Hispanic or Latino	2 (0.7%)
	Neither Hispanic nor Latino	296 (99.3%)
Education	High school or less	93 (32.7%)
	Some college	118 (41.5%)
	Bachelor or higher	73 (25.7%)
Married or live together	Yes	178 (59.7%)
BMI Categories	Underweight (BMI < 18.5)	6 (2%)
	Normal (18.5 ≤ BMI < 25)	97 (32.7%)
	Overweight (25 ≤ BMI < 30)	127 (42.8%)
	Obese (BMI ≥ 30)	67 (22.6%)
Smoking Status	Former Smokers	120 (40.1%)
	Current Smokers	179 (59.9%)
Any family history of lung cancer	Yes	70 (23.4%)
	No	229 (76.6%)
Lung Cancer stage	Early stage (1A-3A)	193 (69.4%)
	Advanced stage (3B-4)	85 (30.6%)
Lesion size of tumor in mm	Mean (SD)	26.3 (20.5)
Lung Cancer Histology	Small cell LC	34 (11.4%)
	Non-small cell LC	265 (88.6%)
Lung Cancer Grade	1 or 2	99 (47.6%)
	3 or 4	109 (52.4%)

Note: Stage information was missing for 21 participants.

**Table 2 ijerph-21-00400-t002:** Univariate Cox proportional hazard regression and lung cancer survival time.

Exposure Variables	OS	PFS
**Tumor characteristics**		
Lung cancer stage: Advanced vs. early stage	7.2 (4.94–10.6), ***p* < 0.0001 ***	7.4 (5.18–10.7), ***p* < 0.0001 ***
Lung cancer histology: small cell vs. non-small cell	3.6 (2.32–5.71), ***p* < 0.0001 ***	3.5 (2.24–5.35), ***p* < 0.0001 ***
Lung cancer grade: 3/4 vs. 1/2	4.2 (2.51–6.91), ***p* < 0.0001 ***	3.3 (2.07–5.18), ***p* < 0.0001 ***
Lesion size in mm (Continuous)	1.03 (1.02–1.04), ***p* < 0.0001 ***	1.03 (1.02–1.04), ***p* < 0.0001 ***
**Personal characteristics/smoking**		
Sex: male vs. female	1.5 (1.06–2.21), ***p* = 0.0231 ***	1.4 (1–1.98), ***p* = 0.0488 ***
Age (continuous)	0.99 (0.961–1.02), *p* = 0.61	0.98 (0.953–1.01), *p* = 0.22
Race: Black vs. White	1.0 (0.46–2.37), *p* = 0.9165	1.0 (0.474–2.17), *p* = 0.9737
BMI: over weight vs. normal weight	0.96 (0.631–1.45), *p* = 0.8353	1.1 (0.726–1.6), *p* = 0.7068
BMI: obese vs. normal weight	1.6 (0.982–2.47), *p* = 0.0597	1.7 (1.09–2.61), ***p* = 0.0191 ***
Any family history of lung cancer: yes vs. no	1.3 (0.849–1.88), *p* = 0.2492	1.3 (0.916–1.92), *p* = 0.1346
Smoking status: current vs. former smokers	1.5 (1.02–2.12), ***p* = 0.0373 ***	1.3 (0.935–1.84), *p* = 0.1165
Smoking duration in years	1.01 (0.99–1.04), *p* = 0.264	1.01 (0.99–1.04), *p* = 0.264
Cigarettes per day (>1 pack vs. ≤1 pack)	1.2 (0.84–1.72), *p* = 0.3124	1.24 (0.89–1.73), *p* = 0.2106
Time to first cigarette (TTFC): ≤5 min vs. longer	1.56 (1.1–2.2), ***p* = 0.013 ***	1.53 (1.1–2.12), ***p* = 0.01 ***
TTFC (4 categories): ≤5 min vs. at least 1 h	2.4 (0.859–6.49), *p* = 0.096	2.8 (1.02–7.63), ***p* = 0.047 ***
TTFC (4 categories): 6–29 min vs. at least 1 h	1.6 (0.57–4.38), *p* = 0.380	1.9 (0.676–5.14), *p* = 0.229
TTFC (4 categories): 30–59 min vs. at least 1 h	1.5 (0.497–4.78), *p* = 0.454	2.1 (0.691–6.28), *p* = 0.193

Note: * *p*-value < 0.05 (i.e., statistically significant).

**Table 3 ijerph-21-00400-t003:** Multivariate Cox proportional regression and lung cancer survival time.

Predictor Variables	OS	PFS
Lung cancer stage: advanced vs. early stage	7.2 (4.81–10.7), ***p* < 0.0001 ***	7.5 (5.11–11.0), ***p* < 0.0001 ***
Sex: male vs. female	1.7 (1.14–2.47), ***p* = 0.0094 ***	1.5 (1.01–2.08), ***p* = 0.0434 ***
BMI: obese vs. normal weight	NA	1.7 (1.09–2.75), ***p* = 0.0209 ***
Smoking status: current vs. former smokers	1.3 (0.911–1.98), *p* = 0.136	1.3 (0.893–1.87), *p* = 0.1736
Time to first cigarette (TTFC): ≤5 min vs. longer	1.0 (0.687–1.46), *p* = 0.991	1.1 (0.753–1.52), *p* = 0.71

Note: * *p*-value < 0.05 (i.e., statistically significant).

**Table 4 ijerph-21-00400-t004:** Causal mediation analysis results for time to first cigarette (X), stage (M), and survival time (Y; overall survival (OS) and progression-free survival (PFS).

	OS			PFS		
	Estimate	95% CI	*p*	Estimate	95% CI	*p*
ACME	−746.47	−1478, −115	**0.02 ***	−563.01	−1135, −68	**0.025 ***
ADE	62.29	−1352, 1594	0.96	−194.27	−1234, 902	0.699
TE	−684.18	−2235, 886	0.38	−757.28	−1923, 410	0.192

Note: ACME: Average causal mediation effect. ADE: Average direct effect. TE: Total effect. * *p*-value < 0.05 (i.e., statistically significant).

## Data Availability

No new data were created from this study.

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
