# Peer review of "Smoking, Lung Cancer Stage, and Prognostic Factors—Findings from the National Lung Screening Trial"

_ijerph, 2024, doi:10.3390/ijerph21040400_

Round 1
Reviewer 1 Report
Comments and Suggestions for Authors
The author assessed the smoking behaviors at lung cancer diagnosis and prognosis using data from the NLST trial. Overall I think it is very well-written. I have a few comments.
1. In your abstract: please provide the full name for TTFC at first appearance.
2. Line 34: HR and CI. Please include full name hazard ratio at first appearance. As well as LDCT below.
3. Line 37: I think the author may have forgotten to add the criteria: smoking at least 20 pack-year as another USPSTF criterion.
4. Line 205: “many late-stage cases are untreated because of the poor prognosis…..” This statement is too general. Is this from the NLST trial or overall from the real-world. Please also add citations.
Reviewer 2 Report
Comments and Suggestions for Authors
Thank you for the possibility of reviewing the submitted manuscript.
General it is well written paper however, I have some doubts.
Minor comments:
1. It would be great if the authors better described the method of randomization.
2. I am wondering if it is correct to write that "The objective was to determine if low-dose computerized tomography (LDCT) reduces mortality from lung cancer compared to standard chest X-ray." The LDCT has no causation in mortality. The implementation of early treatment is correct but no LDCT.
Major comments:
3. the objective of the study should be mentioned at the end of the introduction section, not in the methods section.
4. The conclusions do not cover the objective of the study. They should be objective or conclusions paraphrased.
5. The graphs are not comfortable to read. Could you please improve their quality? Kaplan Meier survival.
Reviewer 3 Report
Comments and Suggestions for Authors
Dear authors,
Your manuscript, "Smoking, lung cancer stage, and prognostic factors - findings from the National Lung Screening Trial", shows results of evaluating the potential participation of the time to the first cigarette (TTFC) metrics on the prognosis of lung cancer patients. Using data from the National Lung Screening Trial, you performed univariate and multivariate Cox models to determine factors that affect the prognosis of these patients. Though the potential contribution of this study to their field, I would like to comment on some concerns:
Major comments
1. In the abstract, please add p-values for HR between (i) current vs. former smokers, and (ii) TTFC<5 min vs. TTFC >5 min.
2. Please add a statement commenting on the statistical power of your sample (n=299). How representative is this cohort from all American individuals with lung cancer that could be potentially included in the study? Which is the acceptable alpha value? Must you consider significant all p-values <0.01? Or does p<0.05 enough?
3. Besides the mean follow-up time, please add the range (minimal-maximal).
4. Please include all significant variables in univariate analysis in the multivariate Cox model.
5. According to Table 3, the TTFC does not directly affect the prognosis of diagnosed patients (p=0.991 for overall survival/OS and 0.71 for Progression-Free Survival/PFS). Then Figure 2 is a bit inconsistent. TTFC is an indicator of nicotine dependence, it could be associated with a higher risk of developing cancer, but the stage at diagnosis seems to be a stronger predictor tool. Please discuss these results properly.
6. Descriptions in lines 224-225 are inconsistent with Table 3. TTFC is not associated with low OS and PFS in the multivariate analysis.
Minor comments
7. Please describe all abbreviations in their first mention (including the abstract). For example, LDCT (abstract).
8. Possible typo: Line 62 "of former..." -> "or former..."
Round 2
Reviewer 3 Report
Comments and Suggestions for Authors
Dear authors,
Your manuscript, "Smoking, lung cancer stage, and prognostic factors - findings from the National Lung Screening Trial", shows results of evaluating the potential participation of the time to the first cigarette (TTFC) metrics on the prognosis of lung cancer patients. Using data from the National Lung Screening Trial, you performed univariate and multivariate Cox models to determine factors that affect the prognosis of these patients. Though the potential contribution of this study to their field, I would like to comment on some concerns:
1. Unfortunately, I see some flaws in this study. The follow-up time is short, which complicates evaluating the impact of TFFC. You could include additional (external) datasets to validate the authors' hypothesis.
2. I will reinforce my suggestion to evaluate the statistical power of your sample (n=299). Applying a post hoc power calculation could help you to determine a reliable alpha value. In their current form, it seems that an alpha value of 0.05 was applied by convenience.
3. Following the second point, lung cancer must be envisaged as a complex disease. Then, multivariate analyses are required to include novel factors in their context. The evaluation of univariate and multivariate analyses suggests that TFFC does not directly affect OS and PFS in lung cancer. Then, it must be properly discussed instead of being overrated
Author Response
1. “I see some flaws in this study. The follow-up time is short.”
The follow-up time for the study is certainly not short. It was planned for 5 years. The NLST is a nationally recognized study that has resulted in 319 publications all based on 5 years of follow-up. https://cdas.cancer.gov/publications/?study=nlst
2. “I will reinforce my suggestion to evaluate the statistical power of your sample (n=299). Applying a post hoc power calculation could help you to determine a reliable alpha value. In their current form, it seems that an alpha value of 0.05 was applied by convenience.”
This is simply fundamentally incorrect. In the power calculation, the alpha value is an input, not an output. The alpha value is set at 0.05 and used to calculate power (1 – β, which is the probability of a true positive. Every NLST publication is based on statistical testing at 0.05. This is basic statistics. Beta is power. Alpha is not. It can even be found on a Wikipedia page (https://en.wikipedia.org/wiki/Power_of_a_test).
3. “Applying a post hoc power calculation could help you to determine a reliable alpha value.”
Even if we assume that power is an alpha calculation, which it is not, post-hoc power calculations are not recommended by statistical theory. I’ve attached a review article that is actually written for journal referees that would be useful for ijerph for future submission. The very premise of the paper is that “post-hoc power estimates should never be requested by reviewers.”
4. “The evaluation of univariate and multivariate analyses suggests that TFFC does not directly affect OS and PFS in lung cancer. Then, it must be properly discussed instead of being overrated.”
But this is exactly what we concluded! There was no direct effect. This was stated. We had graphs, we had tables. It was explained that there is no direct effect. We extensively showed through mediation analysis, that the effect was indirect.